

# Classification of sleep apnea syndrome using the spectrograms of EEG signals and YOLOv8 deep learning model

Kubra Tanci and Mahmut Hekim

Faculty of Engineering and Architecture, Department of Electrical and Electronics Engineering, Tokat Gaziosmanpasa University, Tokat, Turkey

## ABSTRACT

In this study, we focus on classifying sleep apnea syndrome by using the spectrograms obtained from electroencephalogram (EEG) signals taken from polysomnography (PSG) recordings and the You Only Look Once (YOLO) v8 deep learning model. For this aim, the spectrograms of segments obtained from EEG signals with different apnea-hypopnea values (AHI) using a 30-s window function are obtained by short-time Fourier transform (STFT). The spectrograms are used as inputs to the YOLOv8 model to classify sleep apnea syndrome as mild, moderate, severe apnea, and healthy. For four-class classification models, the standard reference level is 25%, assuming equal probabilities for all classes or an equal number of samples in each class. In this context, this information is an important reference point for the validity of our study. Deep learning methods are frequently used for the classification of EEG signals. Although ResNet64 and YOLOv5 give effective results, YOLOv8 stands out with fast processing times and high accuracy. In the existing literature, parameter reduction approaches in four-class EEG classification have not been adequately addressed and there are limitations in this area. This study evaluates the performance of parameter reduction methods in EEG classification using YOLOv8, fills gaps in the existing literature for four-class classification, and reduces the number of parameters of the used models. Studies in the literature have generally classified sleep apnea syndrome as binary (apnea/healthy) and ignored distinctions between apnea severity levels. Furthermore, most of the existing studies have used models with a high number of parameters and have been computationally demanding. In this study, on the other hand, the use of spectrograms is proposed to obtain higher correct classification ratios by using more accurate and faster models. The same classification experiments are reimplemented for widely used ResNet64 and YOLOv5 deep learning models to compare with the success of the proposed model. In the implemented experiments, total correct classification (TCC) ratios are 93.7%, 93%, and 88.2% for YOLOv8, ResNet64, and YOLOv5, respectively. These experiments show that the YOLOv8 model reaches higher success ratios than the ResNet64 and YOLOv5 models. Although the TCC ratios of the YOLOv8 and ResNet64 models are comparable, the YOLOv8 model uses fewer parameters and layers than the others, providing a faster processing time and a higher TCC ratio. The findings of the study make a significant contribution to the current state of the art. As a result, this study gives rise to the idea that the YOLOv8 deep learning model can be used as a new tool for classification of sleep apnea syndrome from EEG signals.

Corresponding author
Kubra Tanci, kubra.tanci@gop.edu.tr

## INTRODUCTION

Sleep apnea syndrome is defined as the interruption of breathing during sleep and the decrease in the amount of oxygen in the blood (*Saha et al., 2016*). There are three different types of sleep apnea syndrome: mixed, central, and obstructive (*Fan et al., 2015*). Central sleep apnea (CSA) is a type of apnea caused by central nervous system-related pauses in the upper airway and lungs during sleep (*Aksahin, Oltu & Karaca, 2018*). Obstructive sleep apnea (OSA), in which the air intake through the nose and mouth decreases and breathing stops during sleep, is the most common type of sleep apnea. Mixed sleep apnea (MSA) initially begins as CSA and later exhibits OSA-like behavior (*Khandoker, Karmakar & Palaniswami, 2011*).

Many studies in the literature have shown that the use of EEG signals provides high correct classification ratios in the detection and classification of sleep apnea. *Alvarez et al. (2009)*, used the entropy of the power spectral density of the EEG signals and oximetric features in the detection of sleep apnea. They achieved a sensitivity of 91%, a specificity of 83.3%, and an accuracy of 88.5% in the detection of sleep apnea. *See & Liang (2011)*, performed sleep apnea diagnosis from EEG signals using a support vector machine (SVM) and achieved an accuracy of 96.2%. *Hsu & Shih (2011)*, used the change of Hilbert spectrum frequency to detect the duration of OSA. In real-time, this system can detect the onset and ending times of sleep respiratory disturbances. *Tagluk & Sezgin (2011)*, calculated quadratic phase coupling (QPC) in frequency subbands by considering the bispectral properties of the EEG signal for the detection of OSA patients and achieved an accuracy of 96.15%. *Aboalayon & Faezipour (2014)*, performed sleep apnea detection with the SVM method by calculating features such as energy, entropy, and standard deviation from the frequency subbands of the EEG signal and achieved an accuracy of 92%. *Schuluter & Conrad (2012)*, proposed an approach for sleep stage scoring and apnea detection using discrete Fourier transform (DFT) and discrete wavelet transform (DWT) with electroencephalogram (EEG), electrocardiography (ECG), electrooculography (EOG) and electromyography (EMG) signals. Success ratios of 95.2% for the sleep apnea scoring and 95.4% for the classification of apnea-hypogene were achieved. *Zhou, Wu & Zeng (2015)*, calculated EEG scaling coefficients using detrended fluctuation analysis (DFA) for sleep apnea diagnosis and achieved the accuracy of 95.1% using SVM. In another study, *Almuhammadi, Aboalayon & Faezipour (2015)*, obtained frequency subbands and used energy and entropy values instead of using the entire EEG signals and achieved a success ratio of 97.14%. *Khalighi et al. (2016)*, developed an automatic sleep stage classification model, and four signal channels (two EEG and two EOG) were selected according to their proximity to the electrode locations of the new model tested in the used dataset. The signals were converted into spectrogram images and classified by using a convolutional neural network (CNN) model, supporting not only its performance on a standard PSG dataset but also the transferability of the model to a dataset measured with the new system. *Chaw, Kamolphiwong & Wongsritrang (2019)*, achieved an accuracy of 91.3085% in sleep

apnea detection using a deep convolutional neural network model based on SPO2 sensor. *Hamnvik, Bernabé & Sen (2020)*, used the You Only Look Once (YOLO) v4 model to diagnose sleep apnea from recordings obtained at a sampling rate of 10 Hz and achieved 71% success. In 2020, *Korkalainen et al. (2020)* performed the classification of sleep stages using a deep learning model. In the three-stage model, 80.1% epoch-epoch accuracy was achieved. In 2021, *Gurrala, Yarlagadda & Koppireddi (2021)* study aims to detect sleep disorders by classifying sleep stages. Using a single EEG channel, the Single Channel Sleep Stage Classification (SS-SSC) method achieved 97.4% accuracy on the Sleep-EDF database. This method, which takes into account both time and frequency features, was tested with an SVM classifier and outperformed previous methods. This approach, which provides high accuracy with single-channel EEG data, makes a unique contribution to the literature. In another study, *Hamnvik (2021)*, used the YOLOv4 model to detect OSA of different lengths and intensities and achieved 87% correct prediction. *He et al. (2022)*, detected sleep apnea using a craniofacial image-based deep learning method. They also reported sensitivity, specificity, and AUC values at various AHI thresholds. *Song et al. (2023)*, developed an OSA screening system based on snoring sound. For this purpose, they developed three models, namely sound features fused with XGBoost, spectrograms fused with CNN model, and spectrograms fused with ResNet, and used them hybridly. They achieved 83.44% accuracy and 85.27% recall. To detect wake states, *Foroughi et al. (2023)*, used ResNet and SVM architectures to reduce computational complexity and simplify feature extraction, achieving an accuracy of 93.82%. *Jo et al. (2023)*, developed models for apnea and sleep stage classification using deep learning methods. *Wang, Koprinska & Jeffries (2023)*, proposed four different deep-learning models to predict the next apnea events in a 30-s segment. Previous studies on the use of time-frequency spectrograms in sleep stage classification have demonstrated the effectiveness of deep learning approaches. In particular, the Time-Frequency Spectra Convolution Neural Network (TFSCNN) proposed by *Jadhav & Mukhopadhyay (2022)* shows promising results in automatic sleep stage scoring. Similarly, *Li et al. (2022)* developed EEGSNet, a deep learning model based on EEG spectrograms. The model was tested on four different data sets: Sleep-EDFX-8, Sleep-EDFX-20, Sleep-EDFX-78 and SHHS.The deep learning method they developed based on EEG spectrograms achieved high accuracy rates in sleep stage classification, reinforcing the effectiveness of such approaches (*Li et al., 2022*).

This study focuses on whether the sleep apnea syndrome is mild, moderate, severe apnea, or a healthy individual by using different apnea-hypopnea index (AHI) values not the detection of the types of sleep apnea syndrome. EEG signal classification is typically performed using raw EEG signals, wavelet transformations, or time-frequency representations. However, EEG spectrograms provide a richer representation by combining both time and frequency information.

This study focuses on a four-class classification problem to demonstrate that spectrogram transformation serves as an effective input for deep learning models. Instead of comparing different transformations, our goal is to evaluate the performance of various architectures in spectrogram-based classification. Therefore, rather than focusing on detailed STFT parameters, we highlight the general effectiveness of spectrograms and the

impact of architectural differences on classification performance. The classification of sleep apnea syndrome from PSG recordings is a time-consuming process due to the overnight recording of patients in sleep laboratories. Classification of sleep apnea syndrome from PSG recordings is time-consuming due to manual analysis of long recordings, often covering the whole night. In our study, we aimed to speed up the analysis process rather than the data collection time. We reduced data complexity and size by converting EEG signals into spectrograms, making the process faster and more efficient. This method minimizes manual steps and provides the time-frequency features of EEG as a single input, making the classification process more efficient. Therefore, in this study, models that classify sleep apnea at different severity levels (mild, moderate, severe, and healthy) using EEG signals are compared. In the literature, there are no studies on the classification of severity levels of sleep apnea syndrome and spectrogram-based classification. This article aims to provide a perspective in this specific area. Metrics such as precision, recall, F1-score, and accuracy are used to comprehensively evaluate the model performance. The YOLOv8 model is analyzed in comparison with ResNet64 and YOLOv5. The analysis shows that YOLOv8 can be more useful in clinical applications with its high success rate and speed advantage. EEG signals providing high classification success ratios are used as the input of ResNet64, YOLOv5, and YOLOv8 models for the classification of sleep apnea syndrome. The research question posed by this study is to what extent the spectrogram of EEG data transformation and subsequent classification is more effective than existing methods. This study aims to compare the performance of different deep learning architectures using the time-frequency features obtained from the spectrogram transformation of EEG signals. It highlights that spectrograms enhance classification accuracy by clearly revealing frequency changes in EEG data over time. The research focuses on identifying the most suitable deep learning architecture for a specific transformation rather than comparing various transformations. Figure 1 shows the schematic illustration of the proposed model.

As shown in Fig. 1, EEG signals from PSG recordings were separated into 30 s segments, which are commonly used in the literature for size equalization, and spectrograms were obtained (*Chaw, Kamolphiwong & Wongsritrang, 2019*). The whole data was divided into training, testing, and validation and first applied as input to the developed ResNet64 model. The same spectrograms were also classified using YOLOv5 and YOLOv8 models respectively and their performance metrics were compared. All experiments were realized on an online cloud-based platform known as Google Colab, which supports the running of codes in Python. The tensor processing unit (TPU) has 35 GB of random-access memory (RAM) and 107.77 GB of storage for data calculations. The rest of the article presents in detail the dataset, materials, and methods used, details of the architecture of the models, performance metrics, experimental results, discussion, and conclusion.

## MATERIALS AND METHODS

### Dataset

The dataset used in this study is C3-A2 channel EEG signals from PSG recordings in the Physionet database (PhysioBank ATM (physionet.org)). These recordings have different



**Figure 1 Schematic illustration of the proposed model.** EEG signals from PSG recordings were separated into 30 s segments and the size was equalized after obtaining the spectrograms. The whole data was divided into training, testing, and validation and first applied as input to the developed ResNet64 model. The same spectrograms were also classified using YOLOv5 and YOLOv8 models respectively and their performance metrics were compared.

AHI values and sleep durations. AHI values are continuous variables measuring the severity of sleep apnea and were used to make the classification more precise. AHI values have been clarified as follows: AHI < 5 (normal), 5 ≤ AHI < 15 (mild), 15 ≤ AHI < 30 (moderate), and AHI ≥ 30 (severe). Although the classification is categorical, AHI values indicate the severity of apnea in each category, which allows for a more detailed analysis of the relationship between EEG signals and sleep apnea severity. The EEG signals used for analysis in this study were selected from patients with different AHI scores and sleep durations, and this information is given in Table 1.

As seen in Table 1, a high AHI value means that the severity of sleep apnea is also high. A value greater than five indicates the severity of sleep apnea syndrome. Sleep apnea syndrome is the breathing disruption for at least 10 s during sleep. EEG signals are divided into 30-s segments covering pre-apnea, apnea, and after-apnea and grouped into four classes: mild, moderate, severe, and healthy (*Tanci & Hekim, 2023*). The night-long data enables a more comprehensive analysis of apnea severity in different sleep stages for the classification of sleep apnea syndrome. This is particularly useful for assessing the variability in the severity and frequency of apnea attacks throughout the night. The difference in sleep durations caused a variance in the number of segments of the signal, so size equalization was performed after obtaining the spectrograms. These spectrograms, which were applied as input to deep learning models, were randomly divided into 70% training, 15% testing, and 15% validation groups. Each spectrogram was generated using 30 s of EEG signal data. In total, 24,000 spectrograms were produced for the analysis. The dataset was divided into 16,800 spectrograms for training and 3,600 spectrograms for testing, with an additional 3,600 spectrograms used for validation. In this study, short-time Fourier transform (STFT) is used to generate spectra from EEG data. Although classical CNN models perform well in classification tasks, they suffer from the vanishing gradient problem. Deep learning models such as Resnet eliminate this problem with its skip connection structure, allowing the model to learn more complex patterns, thus improving classification performance. The ability of the YOLO model to perform inference with a single forward pass allows the model to work quickly and efficiently. Therefore, YOLO has significant advantages, especially in real-time applications and systems requiring low delays. The YOLOv8 model, which we used in this study, contains fewer parameters with its optimized architecture, which reduces the training time of the model and speeds up the inference process. In addition, its lighter weight reduces hardware requirements.

**Table 1 Class, AHI value and sleep durations of selected EEG signals.** A high AHI value means that the severity of sleep apnea is also high. A value greater than five indicates the severity of sleep apnea syndrome. These EEG signals of approximately 6–7 h were segmented into 30 s segments, which is the length frequently used in the literature, and grouped into four classes: mild, moderate, severe.

| Class | AHI value | Sleep duration (h) |
|---|---|---|
| Healthy | 2 | 6.8 |
| Mild | 5 | 6.4 |
| Moderate | 25 | 7.2 |
| Severe | 91 | 5.9 |

## Spectrogram

The spectrogram means the frequencies where the energy of the signal is maximum. The waveform of the EEG signals is two-dimensional while obtaining the spectrogram by adding the frequency content means moving it to three dimensions. In other words, a spectrogram shows the energy change of a signal over time (*Coskun & Istanbullu, 2012*). The spectrogram of a signal is defined as the power distribution of the STFT (*Koseoglu & Uyanik, 2023*). For the STFT application, the moving window function $g(t - \tau)$ is applied to the signal $x(t)$ at time $\tau$. Each window is moved by $\tau$ in the time domain and these changes in the time interval are displayed in the windows. STFT is defined as shown in Eq. (1) (*Foroughi et al., 2023*).

$$X(\tau, f) = \int_{-\infty}^{\infty} x(t)g(t - \tau)e^{-j2\pi ft} dt \tag{1}$$

$x(t)$: *Analized signal*
$g(t - \tau)$: *Windowing function*
$\tau$: *Time-shifting parameter*
$f$: *Frequency parameter*
$e^{-j2\pi ft}$: *The complex exponential function used in the Fourier transform.*

This integral allows the signal to be analyzed over a given time interval ($\tau$) and at a given frequency ($f$). These analysis methods are used to study the frequency components of the signal as a function of time. When $g(t - \tau)$ is considered a windowing function, Eq. (2) is the definition of the STFT. STFT analyzes both the time and frequency information of a signal simultaneously. An example of EEG signals and their spectrograms for the four classes used in the study is given in Fig. 2.

The four spectrograms presented in Fig. 2 were selected to compare and analyze visually different examples of each class in the study. These spectrograms are from 30-s time slices containing representative samples from each class. In this way, the differences and patterns between the classes can be examined more clearly as visual representations.

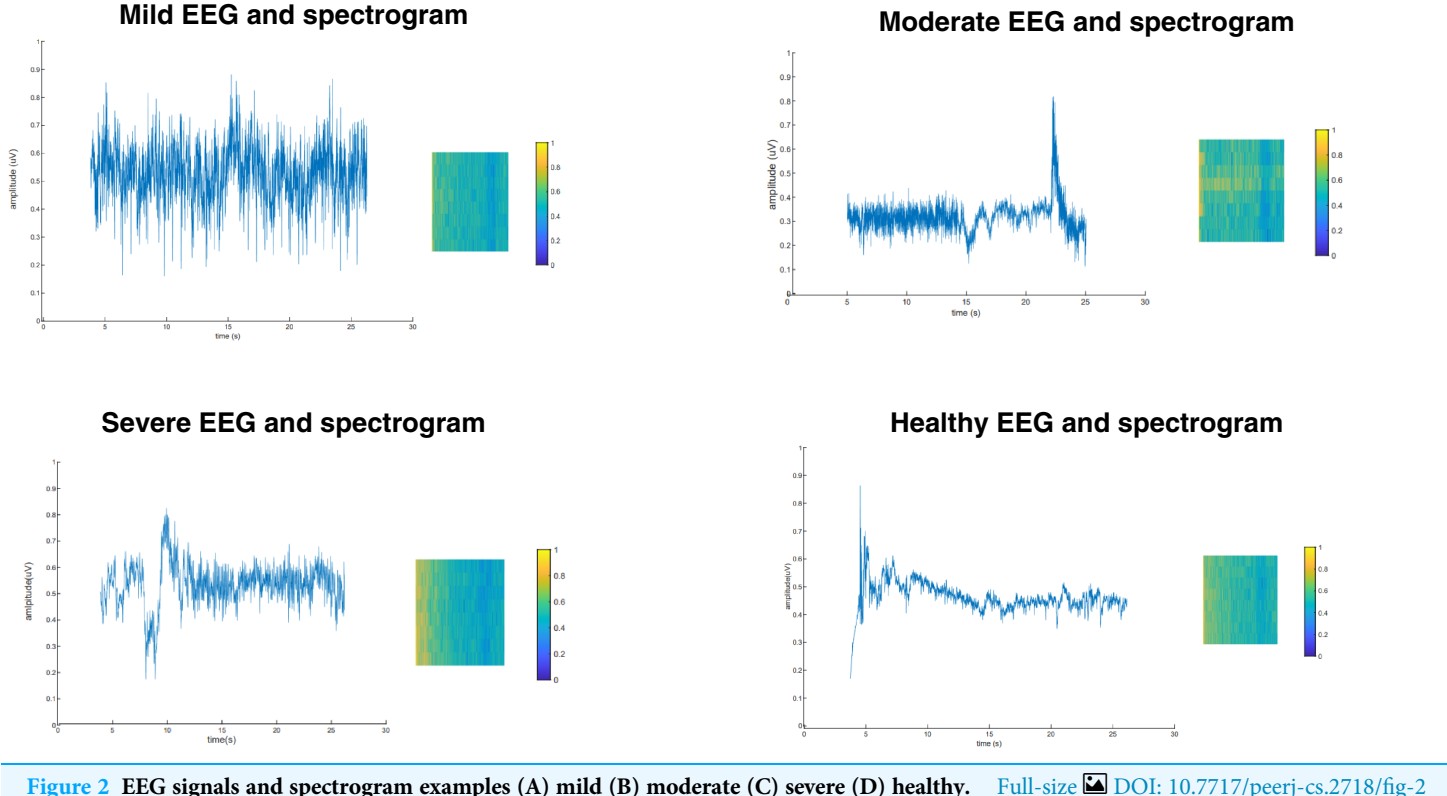

**Figure 2** EEG signals and spectrogram examples (A) mild (B) moderate (C) severe (D) healthy.

## Residual neural networks architecture

The residual neural networks (ResNet) architecture is based on the layer-by-layer combination of residual blocks and enables the training of deeper architectures. In this architecture, blocks with multiple layers are used to reduce the training error, and residual values are added to subsequent layers to learn the difference between input and output. In this case, this prevents lost gradients and makes the network deeper (*Song et al., 2023*). ResNet uses identity mapping to increase the learning capacity and prevent overlearning despite the deepening of the network. The main feature of residual blocks is to equalize the input and output dimensions, thus making it easier for the network to learn. A basic ResNet architecture includes a residual block, two convolutional (Conv) layers, Batch normalization and ReLU activation after each convolution layer, a Residual connection where the input is directly added to the output, and the identity mapping layer where the input and output dimensions are mapped. ResNet models are often referred to by the number of layers. Each additional layer allows the model to learn more complex features. The ResNet model developed in this study is called ResNet64 since it initially contains 64 filters, and the properties of the layers are given in Table 2.

ResNet models have advantages in deep networks such as training simplicity, network augmentation, transfer learning capability, and backward adaptability.

**Table 2 Layers and properties of ResNet64 architecture.** ResNet models have advantages in deep networks such as training simplicity, network augmentation, transfer learning capability, and backward adaptability.

| Layer | Output size | Filters | Activation | Params |
|---|---|---|---|---|
| Conv2d-1 | [batch_size, 64, 112, 112] | 64 | – | 9,408 |
| BatchNorm2d-2 | [batch_size, 64, 112, 112] | 64 | – | 128 |
| ReLU-3 | [batch_size, 64, 112, 112] | – | ReLU (inplace=True) | 0 |
| MaxPool2d-4 | [batch_size, 64, 56, 56] | – | – | 0 |
| Conv2d-5 | [batch_size, 64, 56, 56] | 64 | – | 4,096 |
| BatchNorm2d-6 | [batch_size, 64, 56, 56] | 64 | – | 128 |
| ReLU-7 | [batch_size, 64, 56, 56] | – | ReLU | 0 |
| Conv2d-8 | [batch_size, 64, 56, 56] | 64 | – | 36,864 |
| BatchNorm2d-9 | [batch_size, 64, 56, 56] | 64 | – | 128 |
| ReLU-10 | [batch_size, 64, 56, 56] | – | ReLU | 0 |
| Conv2d-11 | [batch_size, 256, 56, 56] | 256 | – | 16,384 |
| BatchNorm2d-12 | [batch_size, 256, 56, 56] | 256 | – | 512 |
| Conv2d-13 | [batch_size, 256, 56, 56] | 256 | – | 16,384 |
| BatchNorm2d-14 | [batch_size, 256, 56, 56] | 256 | – | 512 |
| ReLU-15 | [batch_size, 256, 56, 56] | – | ReLU | 0 |
| Bottleneck-16 | [batch_size, 256, 56, 56] | – | – | 0 |
| Conv2d-17 | [batch_size, 64, 56, 56] | 64 | – | 16,384 |
| BatchNorm2d-18 | [batch_size, 64, 56, 56] | 64 | – | 128 |
| ReLU-19 | [batch_size, 64, 56, 56] | – | ReLU | 0 |
| … | … | … | … | … |
| AdaptiveAvgPool2d-173 | [batch_size, 2048, 1, 1] | – | – | 0 |
| Linear-174 | [batch_size, 4] | 4 | – | Varies |
| ResNet-175 | [batch_size, 4] | – | – | 0 |

## YOLOv5 architecture

YOLOv5 is a model for object detection and classification and is mainly based on the PyTorch library. It is an architecture consisting of three basic parts: spine, neck, and head (*Fang et al., 2021*). The architecture of the YOLOv5 model is given in Fig. 3.

As shown in Fig. 3, the input images are first transmitted to the backbone network. The feature maps created in this section are combined with different maps. In YOLOv5, Cross Stage Partial Network (CSPNet) is integrated into Darknet and a structure called CSPDarknet is created. CSPNet includes several conv layers, four CSPs with three convolutions, and one spatial pyramid pooling (SPP). The SPP module eliminates the fixed size limitation in the network, so there is no need to denoise, enlarge, and refract images. CSP is a module that compiles features learned in the backbone, making the path between the lower and upper layers shorter. The neck section is known as PANet and it takes all the features extracted from the backbone, saves them, and sends them to the deep layer. These saved features are used as classification labels in the head. In the head layer, convolutional
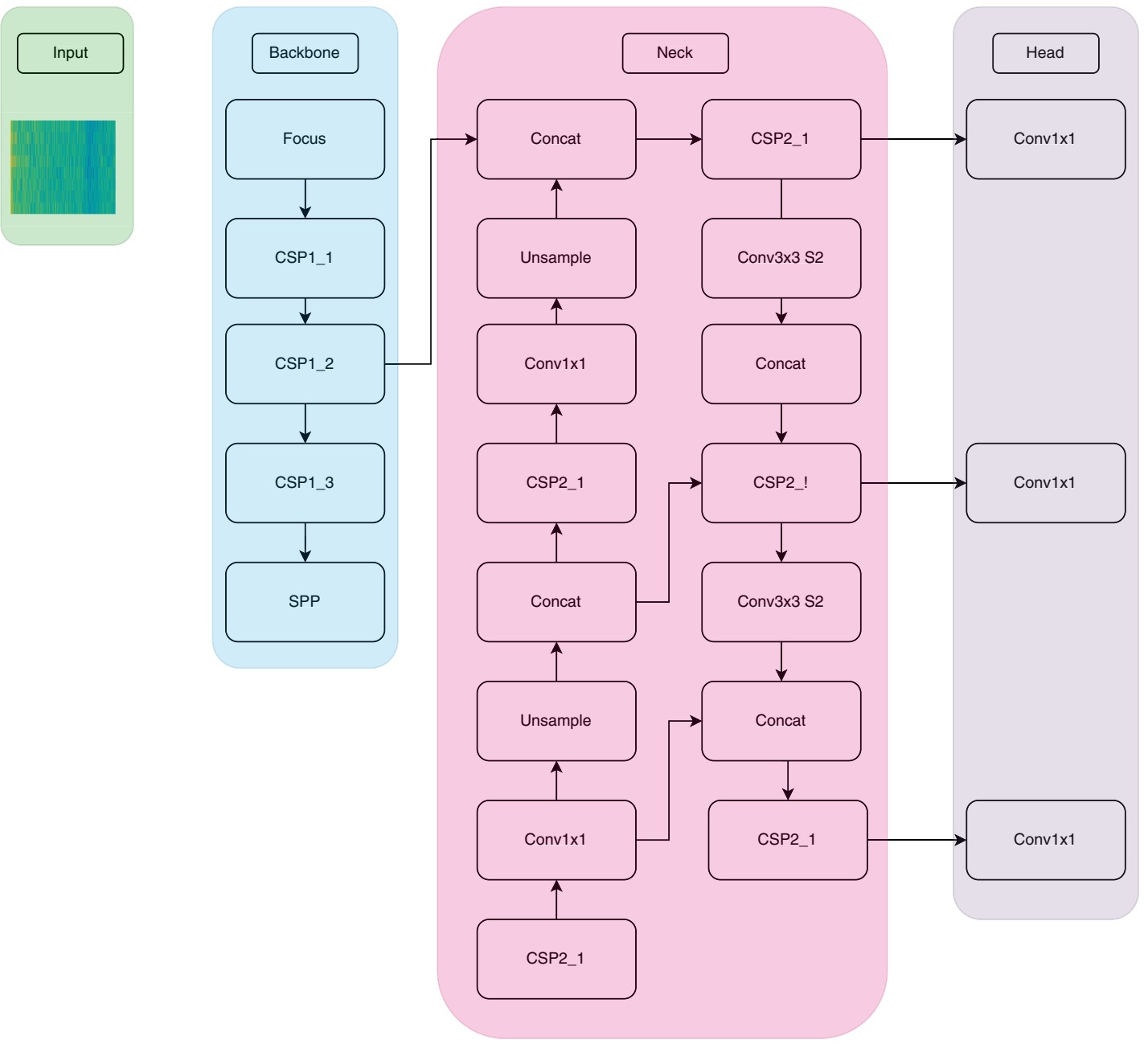

**Figure 3 YOLOv5 architecture.** YOLOv5 architecture consists of backbone neck and head parts.

operations are applied to generate class prediction values based on the target values (*Solawetz & Francesco, 2023*).

## YOLOV8 architecture

YOLOv8 is the latest version of the YOLO family of object detection and classification models. YOLOv8 is an improved version of YOLOv5, adding more advanced features and optimizations to improve its performance. The YOLOv8 architecture has three main

components: backbone, neck, and detection head (*Lou et al., 2023*). The YOLOv8 architecture structure is given in Fig. 4.

As can be seen in Fig. 4, this architecture is based on CSPDarknet53 and includes innovations such as the C2f module, SPPF layer, group normalization, and SiLU activation. There is also a head model that separately handles object presence, classification, and regression functions (*Sary, Andromeda & Armin, 2023*). YOLOv8 computes object detection and class probabilities using sigmoid and softmax functions in the output layer. It uses CIoU, DFL, and binary cross-entropy methods for bounding box and classification losses to improve detection accuracy (*Lou et al., 2023*). The backbone is used to extract features from the input image. CSPDarknet53 effectively reduces the model's parameter count compared to its predecessors, while simultaneously enhancing information exchange between layers. Neck combines the features extracted by the backbone and allows YOLOv8 to detect objects at different scales and aspect ratios. The detection head predicts the position and class of objects in the input image and affects the model's performance by assigning higher weights for hard-to-classify instances. In the post-processing stage, the position and class of objects estimated by the detection head are processed to eliminate low-reliability and overlapping detections. Moreover, data augmentation methods, including random cropping, resizing, and projection, are implemented to augment the variety within the training data, thereby enhancing the model's ability to generalize (*Alsamurai, 2023*).

## Performance assessment metrics

In the study, assessment metrics such as accuracy (Acc), precision (P), recall (R), and F1 score were utilized to evaluate the performance of sleep apnea syndrome classification. Accuracy represents the overall correct classification rate and indicates increasing success as the value approaches 1. Precision (P) denotes the ratio of correctly predicted positive observations to the total predicted positives, while recall (R) signifies the ratio of correctly predicted positive observations to all actual positives. The F1 score is the harmonic mean of sensitivity and precision (*Cecen & Ozer, 2023*). During the calculation of these metrics, the number of true and false predictions is determined using the confusion matrix. The confusion matrix for classification is presented in Table 3.

The metrics to be used in evaluating the performance of the model are computed by using the actual and predicted values of each class. In this study, true and false represent the predictions of the model and their agreement with the actual situation. True positive (TP) represents the correct classification of apnea cases as apnea and true negative (TN) represents the correct prediction of non-apnea cases as non-apnea. In contrast, false positive (FP) represents a non-apnea case being incorrectly classified as apnea and false negative (FN) represents an apnea case being incorrectly predicted as non-apnea. The expressions in the confusion matrix and the performance evaluation metrics are given in Eqs. (2)–(5) (*Tanci & Hekim, 2023*).

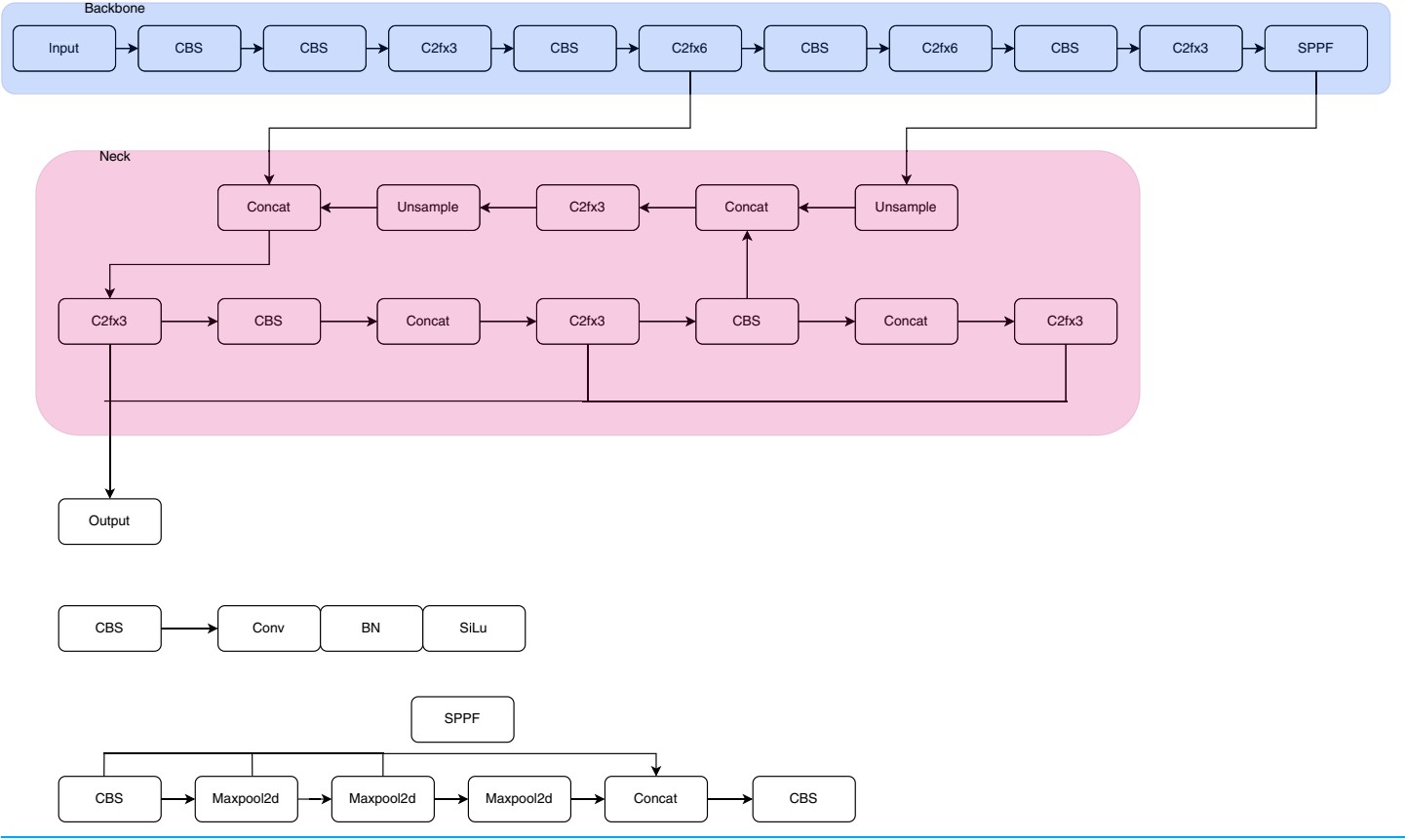

**Figure 4 YOLOv8 architecture.** YOLOv8 architecture is an improved version of the yolov5 model and consists of backbone neck and head parts.

$$Acc = \frac{TP + TN}{TP + FN + FP + TN} \quad (2)$$

$$P = \frac{TP}{TP + FP} \quad (3)$$

$$R = \frac{TP}{TP + FN} \quad (4)$$

$$F1\ Score = \frac{2 * (P * R)}{P + R} \quad (5)$$

TP: true positive, apnea correctly detected as apnea
TN: true negative, non-apnea correctly detected as non-apnea
FP: false positive, non-apnea incorrectly detected as apnea
FN: false negative, apnea incorrectly detected as non-apnea.

## RESULTS AND DISCUSSION

In this study, spectrograms of EEG signals from PSG recordings were utilized as input for sleep apnea classification in ResNet64, YOLOv5, and YOLOv8 models. The input images

**Table 3 Confusion matrix for binary classification.** During the calculation of the metrics, the actual and estimated true/false numbers are determined using the confusion matrix.

|  | Predicted:Mild | Predicted:Moderate | Predicted:Severe | Predicted:Healthy |
|---|---|---|---|---|
| Actual:Mild | TP | FN | FN | FN |
| Actual:Moderate | FP | TP | FN | FN |
| Actual:Severe | FP | FP | TP | FN |
| Actual:Healthy | FP | FP | FP | TP |

were resized to 224 × 224, and training was conducted for 20 epochs for maximum accuracy. The training of the model was limited to only 20 epochs because observations showed that the model performed well enough in the early stages and the risk of overfitting increased as the training continued with more epochs. Therefore, it was decided to limit the number of epochs instead of increasing the training data. Over-learning can negatively impact classification performance on new and unprecedented data, reducing the ability to generalize. In this context, keeping the training time at an optimal level is a critical step to ensure the best validation accuracy. The training results are presented in Table 4.

As seen in Table 4, the number of layers, number of parameters, computational load (GFLOPs), accuracy rate (Top1), and training times of ResNet64, YOLOv5, and YOLOv8 models are compared. GFLOPs are usually short for "Giga Floating Point Operations Per Second" and are used to measure the processing load or computational complexity of a model. That is the intensity of the operations the model performs during training or when making a prediction. Top1 refers to the model's accuracy rate and indicates the proportion of the model's predictions that fall into the correct class with the highest probability. These values are important metrics for evaluating the performance and computational requirements of a model. The number of layers in the YOLOv8 model is 99, while ResNet64 has 175 layers and YOLOv5 has 214 layers. This shows that the number of layers has been reduced by 56% in the YOLOv8 model. In terms of the number of parameters, YOLOv8 has 2,719,288 parameters, while ResNet64 has 25,557,032 and YOLOv5 has 7,030,417 parameters. Comparisons based on these values show that the number of parameters has been reduced by approximately 10.65%. In terms of computational load (GFLOPs), the YOLOv8 model has a value of 4.4 GFLOPs, while the other models have much higher values of 51.1 GFLOPs and 16.0 GFLOPs, respectively. This shows that the YOLOv8 model runs with much less computational power. In terms of accuracy, the YOLOv8 model shows the best performance with an accuracy of 93.7%. In terms of training time, the YOLOv8 model completed the training in the shortest time with 0.139 h and provided an advantage in terms of training. These results show that the YOLOv8 model is both computationally more efficient and an effective option in terms of training time and accuracy. These reductions were made to make the model run faster, less complex, and with less computational effort. Figure 5 illustrates the training and test losses for all three models.

**Table 4 Training results.** The YOLOv5 model has the highest number of layers, whereas the ResNet64 model contains the most parameters.

| Model | Layer | Parameters | GFLOPs | Top1 | Time (h) |
|---|---|---|---|---|---|
| ResNet64 | 175 | 25,557,032 | 0.0511 | 0.930 | 0.174 |
| YOLOv5 | 214 | 7,030,417 | 16.0 | 0.882 | 0.304 |
| YOLOv8 | 99 | 27,192,88 | 4.4 | 0.937 | 0.139 |

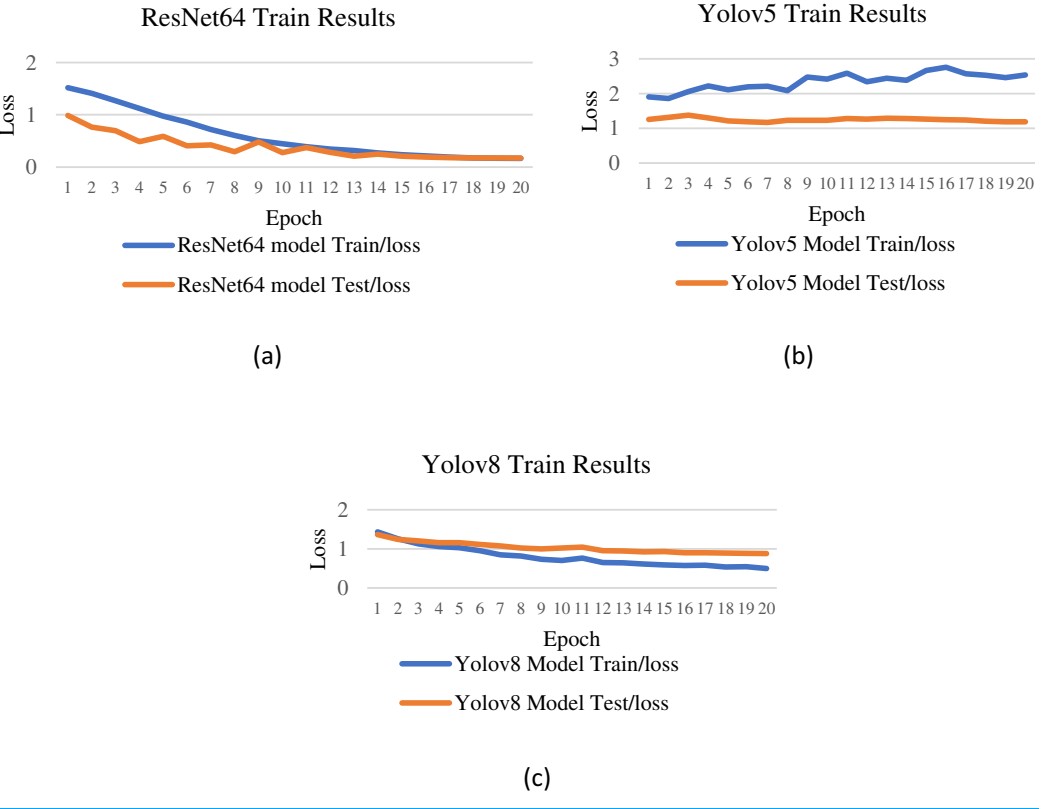

(a)

(b)

(c)

**Figure 5 Test/loss and train/loss results (A) ResNet64 (B) YOLOv5 (C) YOLOv8.** The training loss of the YOLOv5 model shows an upward graph, while this loss gradually approaches 0 in the ResNet64 and YOLOv8 models. This metric measures how far the predictions are from the actual values.

As can be seen in Fig. 5, the training loss of the YOLOv5 model shows an upward graph, while this loss gradually approaches 0 in the ResNet64 and YOLOv8 models. This metric measures how far the predictions are from the actual values. A decreasing train/loss value indicates that the model learns better. As for test/loss, the YOLOv5 model is close to the steady state, while the ResNet64 and YOLOv8 models show a decreasing trend. Furthermore, the performance evaluation metrics of these models are presented in Fig. 6.

As seen in Fig. 6, the lowest recall values for each model were obtained in the healthy class, while the highest recall value in all three models belongs to the severe class. Although the YOLOv8 model achieved the highest value in the mild and moderate classes, both ResNet64 and YOLOV8 models demonstrated equal values in the severe and healthy

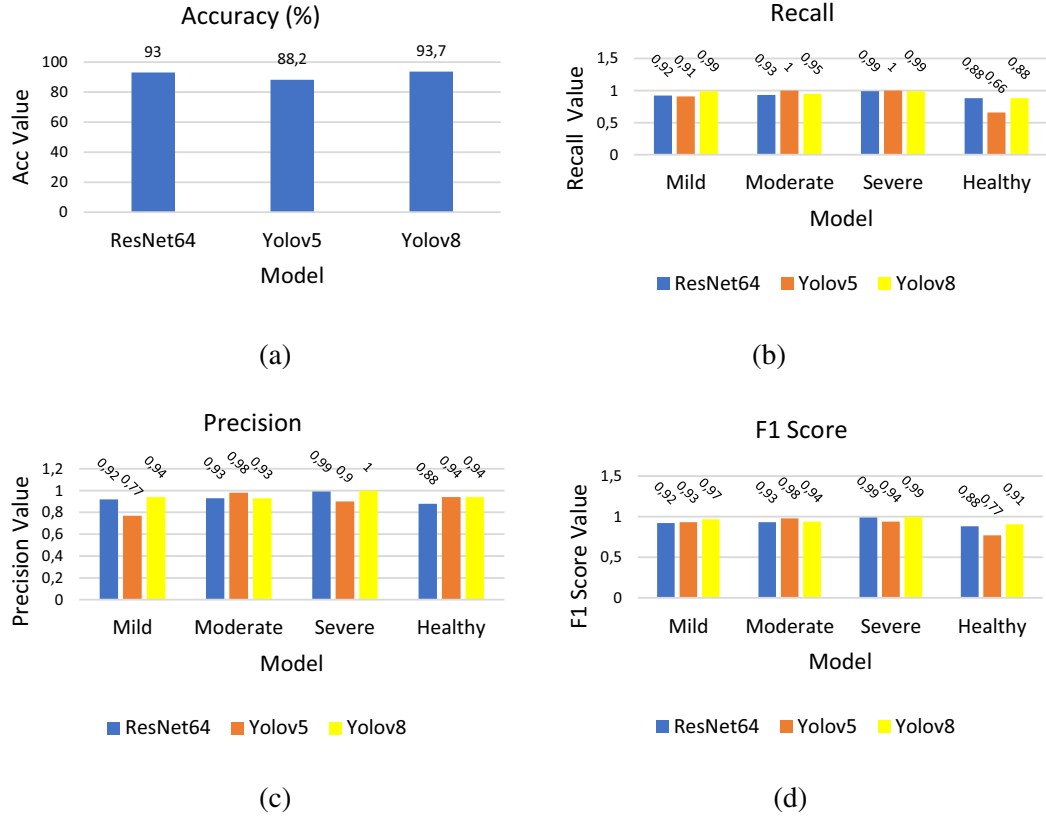

**Figure 6 Acc, recall, precision and F1 scores of three models.** (A) Accuracy (B) recall (C) precision (D) F1 score. The lowest recall values for each model were obtained in the healthy class, while the highest recall value in all three models belongs to the severe class.

classes and outperformed the YOLOv5 model. The precision value is highest in the severe class, with the YOLOv8 model attaining the highest value in the mild and severe classes, and the YOLOv5 model reaching the highest value in the moderate class. In the healthy class, both the YOLOv5 and YOLOv8 models exhibited equal values, which were higher than those of the ResNet64 model.

The F1 score value is consistent with recall and precision values. The YOLOv8 model had the highest value in the mild, moderate, and healthy classes, while in the severe class, both the ResNet64 and YOLOv8 models surpassed the YOLOv5 model. The confusion matrices obtained from the test results for these four groups classified in the study are presented in Table 5.

As seen in Table 5, the highest correct classification rate in all three models belongs to the severe class. In these matrices, rows are actual values and columns are predicted values. In the ResNet64 model, there were 92 correct predictions for the mild class, with one incorrect prediction classified as moderate and seven as healthy. For the moderate class, there were 93 correct predictions, along with one mild, four healthy, and two severe incorrect predictions. Regarding the severe class, there were 99 correct predictions and one incorrect prediction classified as moderate. Lastly, for the healthy class, there were 89 correct predictions, with seven mild and five moderate incorrect predictions. Although the

**Table 5 Confusion matrices of models (A) ResNet64 (B) YOLOv5 (C) YOLOv8.** The highest correct classification rate in all three models belongs to the severe class. In these matrices, rows are actual values and columns are predicted values.

**A**

| ResNet64 | Mild | Moderate | Severe | Healthy |
|----------|------|----------|--------|---------|
| Mild | 92 | 1 | 0 | 7 |
| Moderate | 1 | 93 | 2 | 4 |
| Severe | 0 | 1 | 99 | 0 |
| Healthy | 7 | 5 | 0 | 89 |

**B**

| YOLOv5 | Mild | Moderate | Severe | Healthy |
|--------|------|----------|--------|---------|
| Mild | 91 | 0 | 0 | 27 |
| Moderate | 1 | 100 | 0 | 1 |
| Severe | 4 | 0 | 100 | 6 |
| Healthy | 4 | 0 | 0 | 66 |

**C**

| YOLOv8 | Mild | Moderate | Severe | Healthy |
|--------|------|----------|--------|---------|
| Mild | 99 | 0 | 0 | 5 |
| Moderate | 0 | 95 | 1 | 1 |
| Severe | 0 | 0 | 99 | 6 |
| Healthy | 1 | 5 | 0 | 88 |

number of incorrect predictions for the healthy class is slightly high, the overall accuracy value of the ResNet64 model is 93%. Upon examining the YOLOv5 model, there appear to be more errors in the predictions of the mild class. Additionally, there is an increase in the number of incorrect predictions in the severe class, particularly in the mild and healthy classes. Looking at the confusion matrix of the YOLOv8 model, the performance seems quite good. Most predictions for each class are correct, but as with all models, the number of incorrect predictions in the healthy class is higher than in the other classes. The ROC curves obtained for the models according to these values are presented in Fig. 7.

As seen in Fig. 7, the ResNet64 model achieved the highest correct classification rate in the severe class, followed by moderate, mild, and healthy classes, respectively. In contrast, the YOLOv5 model exhibited the highest correct classification rate in the moderate class, followed by the healthy, severe, and mild classes. Similarly, the YOLOv8 model demonstrated high success rates across all classes, with the highest value observed in the severe apnea class, followed by mild, moderate, and healthy classifications. The area under the ROC curve indicates the level of classification success, with the high accuracy value of severe apnea proportional to the high AHI value.

These experiments show that the YOLOv8 model reaches higher success ratios than the ResNet64 and YOLOv5 models. Although the TCC ratios of the YOLOv8 and ResNet64 models are comparable, the YOLOv8 model uses fewer parameters and layers than the

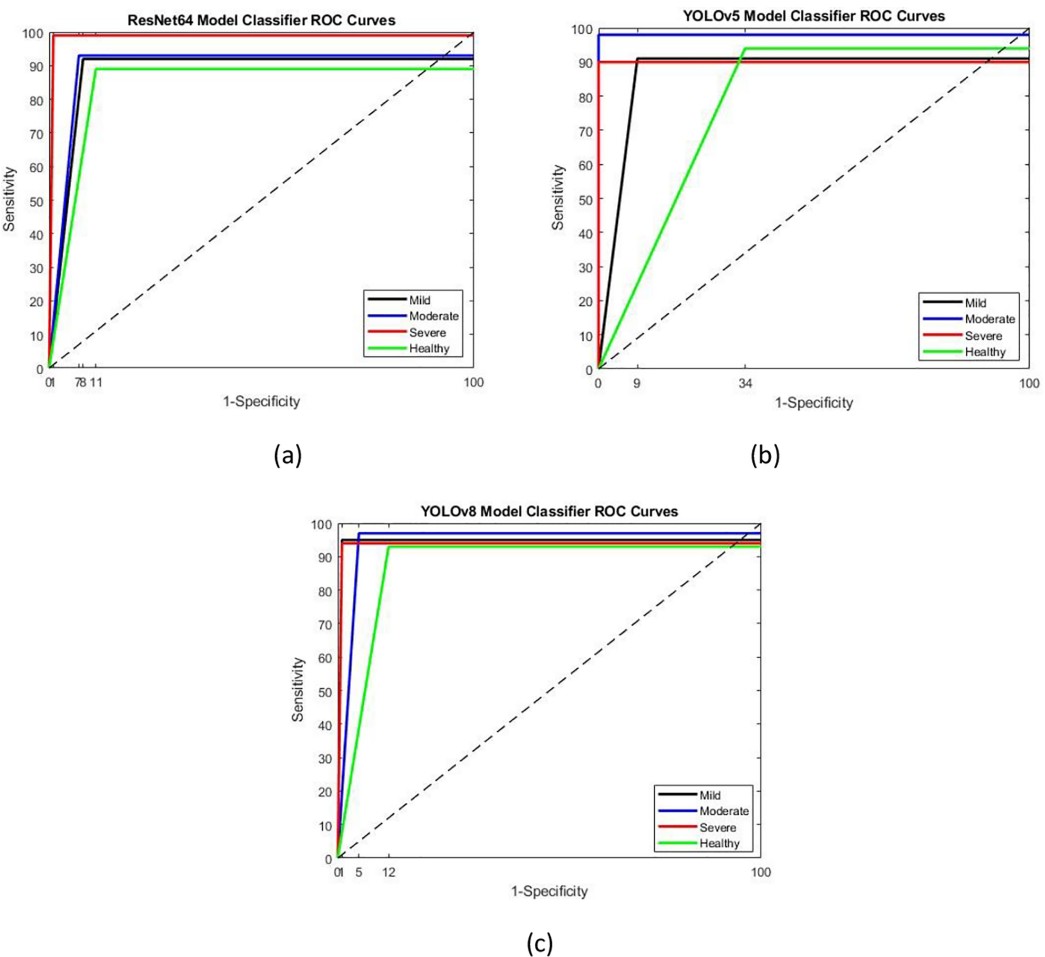

**Figure 7 ROC curves of models (A) ResNet64 (B) YOLOv5 (C) YOLOv8.** The ResNet64 model achieved the highest correct classification rate in the severe class, followed by moderate, mild, and healthy classes, respectively.               

others, providing a faster processing time and a higher TCC ratio. The findings of the study make a significant contribution to the current state of the art. YOLOv8's processing speed and efficiency enhance its usability in time-sensitive situations like sleep apnea. While the ResNet64 model is capable of learning complex patterns due to its deep architecture and residual connections, the lighter and faster structure of YOLOv8 has proven to be a more effective option by reducing hardware requirements during the classification process. The advantages of both models are highlighted, and the critical importance of obtaining quick results in biomedical diagnosis processes is emphasized. The obtained results demonstrate that it is possible to classify sleep apnea using only EEG signals, offering a new approach to the literature. It is thought that this study will inspire future research, and that model performance can be further enhanced by expanding databases with the addition of different EEG channels. The combination of speed, accuracy, and efficiency offered by YOLOv8 makes it a viable alternative for widespread use in sleep apnea diagnosis in clinical settings.

# CONCLUSIONS

In this study, ResNet64, YOLOv5, and YOLOv8 models are proposed for sleep apnea classification from EEG signals. For sleep apnea syndrome classification, EEG signals, which are divided into four classes mild, moderate, severe, and healthy, are segmented into 30-s segments, and spectrograms are taken, resized, and applied as input to all three models separately. The Acc value was 93% in the ResNet64 model, 88.2% in the YOLOv5 model, and 93.2% was achieved in the YOLOv8 model.

In conclusion, due to its high performance, the YOLOv8 model can be used as a new tool for sleep apnea classification from EEG signals. It is thought that the classification of sleep apnea syndrome using only EEG signals without the need for overnight recording in sleep laboratories, and the fact that this proposed model has not been used in previous studies will have an important impact on biomedical applications.

# ACKNOWLEDGEMENTS

This manuscript was partially edited with the assistance of a generative AI tool (ChatGPT).

## Funding
The authors received no funding for this work.

## Competing Interests
The authors declare that they have no competing interests.

## Author Contributions
- Kubra Tanci conceived and designed the experiments, performed the experiments, analyzed the data, performed the computation work, prepared figures and/or tables, authored or reviewed drafts of the article, and approved the final draft.
- Mahmut Hekim performed the computation work, authored or reviewed drafts of the article, and approved the final draft.

## Data Availability
The raw data is available in the Supplemental Files.

The UCD Sleep Apnea Database (ucddb) is available at https://archive.physionet.org/cgi-bin/atm/ATM. Please select the correct database from the dropdown menu on the PhysioBank ATM homepage.

## Supplemental Information
Supplemental information for this article can be found online at http://dx.doi.org/10.7717/peerj-cs.2718#supplemental-information.

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
