# Peer review of "Classification of sleep apnea syndrome using the spectrograms of EEG signals and YOLOv8 deep learning model"

_PeerJ Computer Science, doi:10.7717/peerj-cs.2718_

## Round 0.1 · original submission · Major Revisions

While the topic is worth exploring, the Reviewers agree that further work should be done to ensure that the claim made by the Authors are sustained by the results.

The Authors need to follow carefully the Reviewers’ comments and in particular:

- Provide clear research questions, allowing to understand the novelty of the proposed work.
- Clarify the characteristics of the investigated topic, the data preparation, the classification task, and the rationale behind the choice of specific deep models.
- Answer the doubts related to some strong statements made by the Authors by also providing proper literature references.
- Provide the code and comparisons with literature works to ensure that the reported results are sustained by the applied processing steps.
- Improve the quality of the provided images, which should report sufficient information to be correctly interpreted (e.g., unit measures on the axes).

Moreover, please carefully check the manuscript for formatting issues and missing text.
For example, line 117 presents a missing parenthesis (the Authors could also report the permalink to the repository) and at line 124 only three of the four classes are reported.

The Authors need to provide a point-by-point response to each of the Reviewers’ concerns and a highlighted version of their modified manuscript.

·

Basic reporting

The aim of this manuscript is to improve the classification of sleep apnea from EEG recordings using deep learning and a model with a reduced number of parameters w.r.t. other models. The authors compared ResNet-64, Yolo5 and Yolo8 in a 4-class classification problem from a publicly available dataset (Physionet). Despite the relevance of the topic, the authors' claim is not robustly supported by their results, nor a clear research question has been introduced w.r.t. the state of the art.



ABSTRACT
- no clear research question presented, with reference to gaps in the current state of the art
- no chance level reported (4 classes? then, 25%)


INTRO-SOTA
- not very well introduced what characteristics (e.g., intensity or AHI score, duration, frequency) or classes for sleep apnea are detected/classified in SOTA
- no chance level reported in the difference works reported
- real-time vs night-long monitoring
- what does "new wearable model" mean?
- "The classification of sleep apnea syndrome from PSG recordings is a time-consuming process." Why?
- "the size was equalized after obtaining the spectrograms" Not clear
- not clear what is the gap in the literature and what is the motivation (i.e., research question) of this study

Experimental design

METHODS:
- open-source code?
- Table 1: is the reported AHI value the exact number the AHI can only take in each class? A range was expected.
- " 30 s segments, which is the length frequently used in the literature". Cite references.
- Figure 2: how much representative are the 4 spectrograms of each class?
- why to choose ResNet and Yolo (and not, e.g., CNN)?
- line 223: what does "true/false number" mean?

Validity of the findings

RESULTS:
- why "training was conducted for 20 epochs", only?
- Top2 metric seems to be unuseless, since it is 1 for all models
- there is no discussion with respect to the previous literature. This is very critical to support the impact and novelty of this work.
- Table 3 is trivial
- one of the major claim of the work is the reduced computational complexity of the proposed solution. However, Table 4 per se is not convincing to support this claim.

Additional comments

PRESENTATION:
- figures very small (e.g., Figure 2), no labels, not time resolution/duration indicated, no amplitude/power range reported

Reviewer 2 ·

Basic reporting

The paper has good writing and easy to understand with a good background research and proofs.

Experimental design

1. The methodology applied in this paper is easy to understand and robust. Although, providing useful code bases like- Spectrograms Generation codes, multiple model training pipelines, will make it more accessible to verify the results and utilize the method for future investigations.

Validity of the findings

1. Results claimed in this paper look reasonably good, although verification of these results is difficult with provided information only. Therefore, researchers are highly encouraged to provide as much code base (in form of a public github repository) as possible for further review.

Reviewer 3 ·

Basic reporting

This article processes sleep EEG data in 30-second frames, generates the spectrograms using Short-Time Fourier Transform, and then conducts comparative classification experiments using YOLOv8, ResNet64 and YOLOv5 networks. However, there are still the following shortcomings.
First, what are the innovative points of this paper? Is it the conversion of EEG to spectrograms followed by classification, or is it the classification network?
Second, the lack of comparative experiments is not only in comparison with the results of the ResNet64 and YOLOv5 network models; first, it is necessary to clarify the innovative points.
Third, the quality of the images is unclear, especially Figure 2, which is difficult to see.

Experimental design

The experimental design is unreasonable. The current paper only compares the classification performance of these models, which reduces the paper's innovativeness. If the goal is to highlight the method of converting EEG into the spectrograms, the comparison should be between this method and feature-based as well as end-to-end methods.

Validity of the findings

The author believes the YOLOv8 model can be used as a new tool for sleep apnea classification from EEG signals.

Additional comments

No.

---

## Round 0.2 · Major Revisions

I thank the Authors for sending the revised version of their manuscript.

The Reviewers have highlighted some concerns regarding this new version of the manuscript and asked for clarification on some previously raised issues that seems to be currently missing from your work.

Please, consider Reviewer 1 and Reviewer 3 comments, who provided some clear assessment of the currently remaining issues.

In particular, the Authors should
- Answer the Reviewers’ questions.
- Provide a clear introduction, presentation of the methodology, and assessment of the results.
- Carefully prove some of the provided statements.
- Add a comment regarding the possible availability of the code to all the readerships.
- Carefully check the use of a professional and unambiguous English language.

Please, carefully read the Reviewers’ comments and provide a point-by-point response to each one of them.

·

Basic reporting

The authors have solved my previous doubts, only partially. Below, I report the pending weak points to pay attention and provide a more robust answer to.


- Research question: "to what extent spectrogram of EEG data transformation and subsequent classification is more effective than existing methods". To prove this RQ, you should have compared your architecture with different representations/transformations from the original EEG data. However, this is not the case, as the core of the paper is the comparison of different architectures with the same EEG spectrograms. Please, revise this point.

- ABSTRACT: related question with respect to the relevant state of the art to cite in the article.

- What do you mean with "high-parameters"?

- What do you mean with "the standard reference of levels is set as 25%"?


- INTRO/SOTA: previous Point 4 "not very well introduced what characteristics (e.g., intensity or AHI score, duration, frequency) or classes for sleep apnea are detected/classified in SOTA" not properly addressed. I try to rephrase my concern here: I expect the authors report the relevant state of the art for their claim (that EEG spectrograms represent a more effective input for their architecture compared to other types of transformations). However, the authors replied by describing the segmentation method (every 30s), the transformation (general mention of STFT with no parameters values). The only new information is about they set a 4-class classification problem.

- what do you mean with "In 2021, Vijayakumar Gurrala et al., mention single-channel EEG, which could suggest the use of portable headbands". Not clear. Same type of study? same purpose?

- (previous point 8)"The classification of sleep apnea syndrome from PSG recordings is a time-consuming process." Why? Still not clear. The authors' reply is somehow trivial, as time consumption occurs because recordings last for entire nights. However, the focus of the article is not on a more effective hardware/system to fasten the acquisition times, but on the analysis part. Once more, the logic connection between long acquisition times and the more effective use of EEG spectrograms as an input to a neural network architecture is not clear, nor convincing.

Experimental design

- the code has been provided. However, the question is if the code will be made available to every reader. If so, in what form? GitHub repo?

- Table 1: still cannot get the utility of the AHI values if they are simply transforming a categorical feature ("mild", "healthy", ...) in a number (2, 59, ..). Moreover, the authors replied: "…AHI values and sleep durations of the EEG signals selected from each class to be used in the study are given in the Table 1.” The sentence "selected from each class to be used in the study are given in the Table 1." is not clear at all.


- (previous point 14) "Figure 2: how much representative are the 4 spectrograms of each class?" Not properly addressed. The question related to the representativeness of each of those 4 spectrograms as a prototype of the entire class. The authors should provide a good representative of each class.


- (previous point 16 + point 20) TABLE 3 SEEMS TO BE WRONG. You assigned "TP", "TN" etc to each entry of the table. However, the labels depend on the specific class you are considering for the computation of the true positive, true negative, etc.
Please, refer e.g., to Figure 5.1 of this article: https://www.researchgate.net/publication/314116591_Activity_Context_and_Plan_Recognition_with_Computational_Causal_Behaviour_Models
to check your correct understanding of a multi-label confusion matrix. Then, correct the Table and revise your results, if needed.

Validity of the findings

- (previous point 17) why "training was conducted for 20 epochs", only? The authors' reply is not convincing. If overfitting occurs, this can be overcome with increasing the training set or reducing the number of epochs and checking if the validation set classification reaches a satisfactory classification accuracy. Overfitting is not a good point to reach, as this means that any NEW data giving as an input to the network will be most likely mis-classified, if slighlty different from what the network has seen during training.

- (previous point 19) unless "there are no studies on the classification of severity levels of sleep apnea syndrome and spectrogram-based classification", a reference with previous literature or expected results should be added. For example, there are previous studies showing the different performance of models using raw EEG data or spectrograms. For example, this paper (https://www.mdpi.com/1660-4601/19/10/6322) proposed the use of EEG spectrograms in sleep stage scoring and showed the advantages of this transformation over classification of raw EEG samples. The authors need to make themselves in relation to previous knowledge.


- (previous point 21) "The number of layers was reduced by 56% and the number of parameters by about 10.65%." This sentence is not fully clear. The number of layers of what model? the number of parameters of what model? I assume the authors meant the number of layers and parameters of their proposed solution. However, the text is not clear enough in the present form.

Additional comments

Many English language-related errors are still present (e.g., "Accordingly, throughout the manuscript, we have revised."... missing object of the "revision").

Weird terminology is somewhere used: e.g., "the size was equalized after obtaining the spectrograms" and "the standard reference of levels is set as 25%".

Some notation is not specified (e.g., in Table 4, "GFLOPs" and "Top1".

Figure 2 is still very poor in quality and it misses values, both in the time-domain and in the spectra. Did the authors use the same ranges (time and frequency and dB) for all figures? This is relevant to farily compare the four subfigures.

Are Figure 3 and 4 original, drawn by the authors? As YOLOv5 and YOLOv8 are previously published architectures, it is important that a reference to the original papers is added (also to the figures captions).

Reviewer 2 ·

Basic reporting

no comment

Experimental design

no comment

Validity of the findings

no comment

Additional comments

The revised version makes the findings of this paper clear and easy to understand.

Reviewer 3 ·

Basic reporting

point1
The introduction section is somewhat disorganized. Based on the title and the subsequent research content, it should primarily describe what signals have been used by researchers studying sleep-disordered breathing and which algorithms they have employed.
point2
The formula 2 is incorrect; it should be the Discrete Fourier Transform.

Experimental design

point3
The term "TCC" appears a total of four times in the text, with two occurrences in the abstract and two in the results section. What exactly does TCC refer to?
point4
What does AHI refer to in the text? These recordings have different AHI values and sleep durations. A value greater than 5 indicates the severity of sleep apnea syndrome in your manuscript. But
generally speaking: AHI < 5 is considered normal; mild OSA is defined as 5 ≤ AHI < 15; moderate OSA as 15 ≤ AHI < 30; and severe OSA as AHI ≥ 30.
point5
This paper transformed EEG signals into spectrograms for analysis. How long was the duration of EEG signals used to generate a single spectrogram? How many spectrograms were used in the algorithm analysis? What were the sizes of the training and testing sets, respectively?
point6
AHI is the total number of apnea and hypopnea events occurring per hour of sleep. For each image, how was its classification result (severity of sleep apnea) labeled based on the AHI index?

Validity of the findings

no comment

Additional comments

no comment

---

## Round 0.3 · accepted · Accept

I thank the Authors for having addressed all the Reviewers’ comments.
Your manuscript is now ready for publication.

I have thoroughly checked the provided responses and modifications and assessed that:
- the introduction, methodology and results were clearly presented, following the Reviewers’ guidance;
- the unclear statements pointed out by Reviewer 1 were clarified;
- the figures and tables have been checked;
- the acronyms were clarified as per Reviewer 3’s suggestions;
- the AHI values were clearly reported as well as the details on the division of the data in training, validation and test sets;
- the use of the English language has improved.

Please, consider following Reviewer 1’s final comment on your Python code, while the publication process proceeds.

The PeerJ staff will surely help you during the final stage of your submission, helping you in publishing high quality images and tables.

·

Basic reporting

In the new updated version of the manuscript, the authors have addressed all my previous concerns and now the paper shows sufficient quality and robustness to be accepted.

I would further suggest the authors add comments to their Python code, possibly sharing it as a Colab notebook where to add textual sections, much better visible than single-line comments in between many code lines (as in the present form).

Experimental design

properly addressed

Validity of the findings

properly addressed

Additional comments

No additional comments.